# Nevus-Associated and De Novo Melanoma: A Cross-Sectional Study on Prognostic Differences

**DOI:** 10.3390/cancers17233859

**Published:** 2025-11-30

**Authors:** Emi Dika, Federico Venturi, Biagio Scotti, Alberto Gualandi, Carlotta Baraldi, Sabina Vaccari, Sebastiano Posenato, Corrado Zengarini, Aurora Alessandrini, Leonardo Veneziano, Marco Ardigò, Elisabetta Magnaterra

**Affiliations:** 1Department of Medical and Surgical Sciences (DIMEC), University of Bologna, 40126 Bologna, Italyfederico.venturi@hotmail.it (F.V.); corrado.zengarini@studio.unibo.it (C.Z.);; 2Oncologic Dermatology Unit, IRCCS Azienda Ospedaliero Universitaria di Bologna, 40126 Bologna, Italy; 3Dermatology Unit, IRCCS Humanitas Research Hospital, 20089 Rozzano, Italy; marco.ardigo@humanitas.it; 4Department of Biomedical Sciences, Humanitas University, 20072 Pieve Emanuele, Italy

**Keywords:** nevus-associated melanoma, de novo melanoma, melanoma prognosis, melanomagenesis

## Abstract

Melanoma is a serious subtype of tumor of the skin that can develop either from healthy skin or from a pre-existing mole. Understanding whether a melanoma arises on its own or from a mole is important, as this may influence how the tumor behaves and how it should be managed. In this study, we analyzed nearly 400 patients with melanoma to compare nevus associated melanoma and de novo melanoma. We found that melanomas associated with nevi were usually thinner, showed fewer signs of aggressive growth, and were more common in younger patients and on the trunk. However, long-term survival was similar between the two groups. These findings highlight the importance of regular skin checks, especially for people with many moles, and suggest that further research into the biological differences between these melanoma subtypes may help improve prevention and treatment strategies.

## 1. Introduction

Cutaneous melanoma is a malignancy of increasing global incidence and remains a leading cause of skin cancer-related mortality [1,2,3,4]. While the majority of melanomas are believed to arise de novo, a significant subset develops in association with pre-existing melanocytic nevi, a phenomenon that has important implications for both clinical practice and our understanding of melanomagenesis [5,6,7,8,9,10,11,12,13,14,15,16]. The relationship between nevi and melanoma is complex: although only about one-third of primary melanomas are histologically associated with a nevus precursor, the presence of numerous nevi is one of the strongest known risk factors for melanoma development, including for tumors that do not arise directly from nevi [3,5,10,14]. This paradox highlights the multifactorial nature of melanoma pathogenesis, involving genetic predisposition, environmental exposures, and molecular alterations [4,17,18].

Recent large-scale retrospective and population-based studies have provided new insights into the clinicopathologic and molecular distinctions between nevus-associated melanomas (NAM) and de novo melanomas (DNM). NAMs are more frequently diagnosed in younger patients, often present on the trunk, and are associated with a higher total nevus count and a history of numerous moles during adolescence [8,10,12,19,20,21,22]. Histopathologically, NAMs are commonly thinner at diagnosis, less likely to exhibit dermal elastosis, and are often linked to the superficial spreading subtype [14,16,20]. Molecular analyses have revealed a higher prevalence of BRAF V600E mutations in NAMs, supporting the concept of distinct genetic pathways in their development [10,12,14]. Notably, a recent retrospective study found that nearly 80% of melanoma in situ cases (excluding lentigo maligna and acral lentiginous subtypes) were associated with a nevus, and the vast majority of these were dysplastic nevi, particularly those with severe dysplasia [16].

Despite these advances, many questions remain regarding the biological equivalence, prognostic significance, and optimal management of NAM compared to DNM [16,20]. The identification of nevus-associated pathways may have implications for risk stratification, surveillance, and targeted prevention strategies, especially in individuals with high nevus counts or dysplastic nevus syndrome [10,21]. In this context, detailed clinicopathologic characterization of NAM cases is essential to further elucidate their unique features and to inform evidence-based clinical guidelines.

Here, we present our case series of melanoma arising on a nevus, aiming to contribute to the growing body of literature on NAM and to enhance understanding of its clinical and biological significance within the spectrum of cutaneous melanoma.

## 2. Materials and Methods

We conducted a retrospective, single-center study based on an anonymized database derived from patients attending the Melanoma Clinic of the Dermato-Oncology Unit, IRCCS Policlinico Sant’Orsola-Malpighi, Bologna. The study period extended from January 2007 to December 2021. A total of 378 patients with invasive melanoma were included, classified into two cohorts: 288 with DNM, defined as absence of a histologic nevus remnant, and 90 with NAM, defined by histologic continuity with a pre-existing melanocytic nevus.

For each patient, demographic, clinical, and histopathologic variables were recorded, including age at diagnosis, sex, anatomic site, Breslow thickness, mitotic rate, ulceration, sentinel lymph node (SLN) status, lymph-node or distant metastasis, and melanoma-specific mortality. All patients underwent surgical excision with subsequent histopathologic examination. SLN biopsy was performed in 20 NAM cases and 220 DNM cases.

Histopathologic assessment was independently performed by two dermatopathologists. Tumors were classified as NAM when contiguous nevus components (common, congenital, dysplastic, or nevus spilus) were identified, and as DNM when no such component was present. Discrepancies were resolved by consensus.

Continuous variables (age, Breslow thickness, mitotic rate) were analyzed using the Student’s *t*-test for equality of means and Levene’s test for homogeneity of variances. Categorical variables (sex, anatomic site, ulceration, SLN status, presence of a positive SLN, lymph-node or distant metastasis, melanoma-specific mortality) were assessed using Chi-square or Fisher’s exact test as appropriate. A multivariate logistic regression model was applied to evaluate independent associations with NAM classification. Statistical significance was set at a two-sided *p* value < 0.05. Analyses were performed using IBM SPSS Statistics (version 30, IBM Corp., Armonk, NY, USA).

## 3. Results

A total of 378 patients were included in the study cohort: 288 were classified as having DNM and 90 as NAM. The mean age at diagnosis was similar between groups (54 years for DNM vs. 52 years for NAM). A slight male predominance was observed in the NAM group (58%). With regard to anatomical distribution, the trunk was the most commonly affected site in both groups, though significantly more frequent in NAMs (65.6%) compared to DNMs (51.7%). Although a slight male predominance was observed in the NAM cohort, no sex-related differences were identified in either the NAM or DNM groups with respect to anatomical site distribution, age at diagnosis, or any histopathologic or clinical parameters. An overview of the clinical and anatomical characteristics is provided in Table 1.

The histopathologic evaluation of the melanomas revealed important differences between the NAM and DNM groups. NAMs exhibited a significantly lower mean Breslow thickness, measuring 0.55 mm, compared to 0.84 mm in de novo melanomas (Table 1, Figure 1).

This thinner tumor profile in NAM suggests earlier detection or a slower growth trajectory. Similarly, mitotic activity—a key indicator of tumor proliferation—was markedly reduced in NAMs, with a mean of 0.17 mitoses per mm^2^ versus 1.21 in DNMs (Figure 1). Only 2.2% of NAMs presented with ulceration, compared to 9.4% of DNMs. Distant metastases were observed exclusively in patients with DNM (6.6%), while none of the NAM patients developed metastatic disease during follow-up. However, no significant differences were found between the groups regarding SLN positivity or melanoma-specific mortality.

Multivariate logistic regression analysis confirmed that lower Breslow thickness and mitotic rate were independently associated with NAM classification (Table 2).

## 4. Discussion

This study confirms several clinicopathologic distinctions between NAM and DNM, supporting the concept of divergent biological pathways in melanomagenesis. In our cohort, NAM accounted for approximately 24% of invasive melanomas, a prevalence consistent with prior reports ranging from 15% to 30% [12,15,16,21]. Patients with NAM were slightly younger at diagnosis (mean age 52 years vs. 54 years for DNM), and a male predominance was observed in the NAM group (58%). The trunk was the most common site of involvement, significantly more frequent in NAM (65.6% vs. 51.7%). These findings mirror those of Lai et al. [22], who similarly reported younger age and trunk predilection in NAM, supporting the hypothesis that nevus biology and host phenotype contribute to NAM pathogenesis.

Histopathologic features also differentiated the two groups. NAMs were diagnosed at a thinner stage, with a mean Breslow thickness of 0.55 mm compared with 0.84 mm in DNMs, and exhibited lower mitotic activity (0.17 vs. 1.21 mitoses/mm^2^). Ulceration was rare in NAM (2.2%) but more frequent in DNM (9.4%). No distant metastases were observed in NAM during follow-up, whereas 6.6% of DNM patients developed metastatic disease. These results are consistent with prior large-scale studies showing that NAMs present with more favorable histopathologic parameters, including reduced thickness and lower proliferative indices [21,22]. Multivariate analysis confirmed that Breslow thickness and mitotic rate were independently associated with NAM, suggesting that these tumors may either be detected earlier due to clinical surveillance of nevi or represent biologically less aggressive lesions.

Despite these favorable features, we found no significant differences between NAM and DNM in SLN positivity (1.1% vs. 6.3%) or melanoma-specific mortality (0% vs. 0.69%). This aligns with previous literature indicating that while NAMs are often diagnosed at an earlier stage, long-term prognosis is similar to DNMs once adjusted for established prognostic factors [12,15,16,21]. These observations underscore the importance of early detection and targeted surveillance in high-risk individuals, such as those with numerous or atypical nevi, but also highlight that histopathologic advantages at diagnosis do not necessarily translate into survival differences.

Our results also underline the diagnostic challenges in distinguishing NAM from DNM. Tumor overgrowth may obscure nevus remnants, leading to underestimation of NAM prevalence [20,21]. Dermoscopic and histopathologic clues, such as regression or negative pigment network, may suggest NAM but lack specificity, with frequent overlap with benign nevi and DNMs [13,20,21]. Future advances in confocal microscopy, molecular diagnostics, and integrative imaging approaches may help refine classification and improve diagnostic accuracy. Some illustrative examples of the clinical and dermoscopic features of NAMs are provided in Figure 2.

Molecular evidence reinforces the distinctiveness of NAM and DNM. NAMs are more frequently associated with BRAF V600E mutations, high nevus density, and lower dermal elastosis, whereas DNMs are more often linked to cumulative sun damage, older age, and heterogeneous mutation profiles [12,15,16,21,23,24].

Although the complete evolutionary trajectory of melanoma—whether arising de novo or from a pre-existing nevus—remains a matter of debate, the 2023 World Health Organization (WHO) classification of skin tumors redefines distinct evolutionary pathways of melanomagenesis based on genetic alterations, precursor lesions, and the degree of chronic sun damage (CSD) [24,25,26,27]. These include low-CSD melanomas, such as superficial spreading melanoma, which may arise from a nevus or dysplastic nevus, and high-CSD melanomas, such as lentigo maligna melanoma, which typically originate from melanoma in situ rather than benign precursors. Within this framework, NAMs correspond to the low-CSD category and are characterized by a stepwise progression from a benign BRAF V600E-mutated nevus through intermediate lesions harboring TERT, CDKN2A, and SWI/SNF mutations, to in situ and invasive melanoma, with PTEN alterations emerging at later stages [10]. Conversely, the development of DNM appears to be driven by a heterogeneous set of early mutations, including BRAF variants such as V600K, K601E, and G469A, as well as NRAS and NF1 mutations. In these tumors, subsequent progression involves acquisition of the same driver alterations seen in NAM, but frequently accompanied by UV-induced TP53 damage typical of high-CSD melanoma [26,27].

A pivotal study by Shirata et al. [28] further explored the molecular relationship between the nevus and melanoma components in NAM. In their cohort of 60 cases, the majority (78.3%; 47/60) displayed concordant BRAF mutational status in both components (mut/mut or WT/WT), with 48% (23/47) harboring BRAF V600E mutations—consistent with previous reports [21,28,29,30]. Among these concordant pairs, the proportion of BRAF V600E-mutated cells was comparable in melanoma and nevus in 39% (9/23) of cases, whereas in 48% (11/23), the melanoma harbored a higher percentage of mutant cells compared with the adjacent nevus. These findings support the hypothesis that BRAF V600E mutation confers a proliferative or survival advantage promoting melanoma progression.

In a subset of discordant cases (21.7%; 13/60), only one component—either the melanoma or the nevus—was mutated, while the counterpart remained wild-type. Since BRAF mutations are typically fully clonal in melanocytic nevi, cases where only the nevus was mutated may suggest the absence of a direct clonal relationship between the two lesions. Conversely, when the mutation was detected solely in the melanoma, it remains plausible that the tumor arose from a pre-existing nevus in which the mutant portion was overgrown or not available for molecular analysis, or that the mutation was acquired during tumor progression [28].

Recent studies focusing on non-coding RNAs and global gene expression patterns provide additional insight into these divergent biological pathways [28,29,30,31,32,33,34,35,36,37,38,39]. Analyses of small-RNA sequencing in dysplastic and congenital nevi, as well as in melanomas arising from nevi, revealed distinct miRNA expression profiles [28]. Hierarchical clustering and principal component analysis demonstrated that dysplastic nevi clustered closely with their corresponding congenital nevi, but diverged significantly from melanoma samples [28,29,30,31,32,33,34]. These findings suggest that dysplastic nevi possess a specific miRNA signature, distinct from that of melanoma, supporting the notion that they represent separate biological entities rather than obligate precursors of malignancy [28,29,30,31,32,33,34].

Collectively, these molecular and transcriptomic observations strengthen the concept that NAM and DNM represent biologically distinct pathways of melanomagenesis—one following a nevus-to-melanoma progression model and the other developing independently from isolated melanocytes accumulating pathogenic mutations [31,32,33,34]. Future studies integrating genomic, transcriptomic, and epigenetic profiling may help define the molecular continuum between these entities and refine their clinical classification [34,35,36,37,38,39,40,41,42,43,44].

These biological differences are not only of academic interest, but may also translate into clinically meaningful distinctions, particularly in terms of prognosis and treatment response. Breslow thickness, mitotic activity, and ulceration are key prognostic determinants in melanoma and directly influence staging, sentinel lymph node biopsy decisions, and eligibility for adjuvant therapy [45,46,47,48,49,50,51,52]. The consistently higher Breslow thickness, increased mitotic rate, and greater frequency of ulceration seen in DNM align with features associated with poorer prognosis and more aggressive therapeutic recommendations. Conversely, the thinner profile and lower proliferative activity observed in NAM correspond to factors typically linked to earlier-stage disease. Although these differences did not translate into divergent outcomes in our cohort—likely due to the low number of metastatic and fatal events—they remain clinically meaningful within current prognostic frameworks.

Biological distinctions between NAM and DNM may also carry therapeutic implications. NAMs more frequently harbor BRAF V600E mutations, which are predictive of responsiveness to targeted BRAF/MEK inhibition, while DNMs—often arising in chronically sun-damaged skin—exhibit broader mutational spectra and may display different immunologic profiles that influence response to immunotherapy [14,53,54,55,56]. These emerging data reinforce the notion that melanoma subtype may have relevance not only for prognostic assessment but also for treatment selection [57,58,59,60]. Future studies integrating molecular profiling, treatment patterns, and outcomes are needed to determine whether these biological differences translate into clinically distinct therapeutic trajectories.

This study has limitations. Its retrospective design and reliance on histopathologic identification of nevus remnants may underestimate the true prevalence of NAM, since nevus components may be obscured by tumor overgrowth [61,62]. In addition, we did not perform formal survival analyses, such as Kaplan–Meier curves or Cox regression, which limits the ability to draw firm conclusions on long-term prognosis. Indeed, because metastatic and fatal events were extremely infrequent in our cohort (19 metastases and 2 melanoma-specific deaths), outcome-based multivariable models could not be reliably performed. Larger multicenter cohorts will be required to clarify whether melanoma subtype exerts an independent influence on metastatic risk or survival. Moreover, the absence of molecular characterization further restricts a direct comparison of genetic differences between subtypes. Future prospective studies integrating molecular profiling, dermoscopy, and survival data are warranted to provide a more comprehensive understanding of the biological and clinical distinctions between NAM and DNM.

## 5. Conclusions

In summary, our study confirms that NAMs are more likely to occur in younger patients, on the trunk, and are diagnosed at a thinner stage with lower mitotic activity and less ulceration compared to DNMs [12,15,16,21,63,64]. While these features suggest a less aggressive phenotype, long-term outcomes appear similar when adjusted for stage. Enhanced surveillance of nevi in high-risk individuals may facilitate earlier detection of NAM, but further research is needed to clarify the biological and prognostic distinctions between NAM and DNM and to optimize management strategies for both entities.

## Figures and Tables

**Figure 1 cancers-17-03859-f001:**
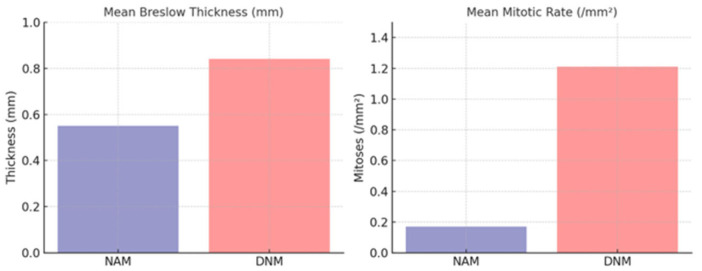
Comparison of mean Breslow thickness (**left**) and mean mitotic rate (**right**) between NAM and DNM groups. NAMs displayed significantly lower values for both parameters.

**Figure 2 cancers-17-03859-f002:**
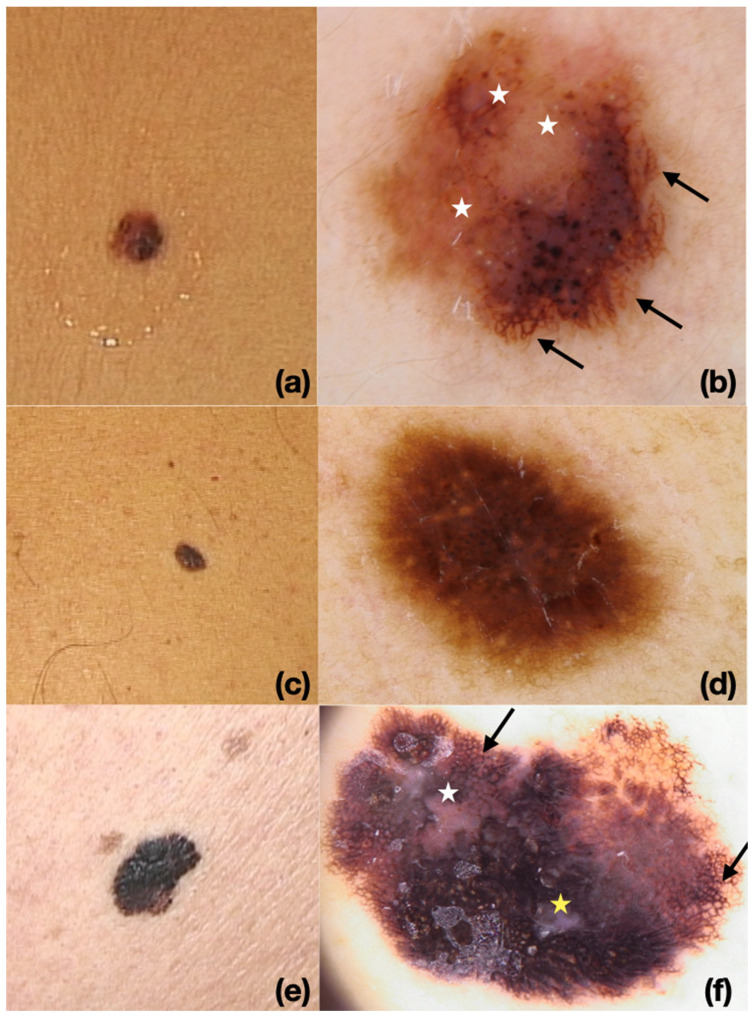
(**a**,**b**) Melanoma arising in a nevus on the back (Breslow thickness: 0.7 mm). (**a**) Clinical image. (**b**) Dermoscopy at 10× showing asymmetric architecture with areas of regression (white star) and atypical pigment network (black arrow). (**c**,**d**) NAM on the trunk (Breslow thickness: 0.3 mm). (**c**) Clinical image. (**d**) Dermoscopy at 10× revealing irregular pigment network. (**e**,**f**) NAM of the back (Breslow thickness: 1.4 mm). (**e**) Clinical image. (**f**) Dermoscopy at 10× showing asymmetric, multicolored pattern with regression areas (white star), a blue-white veil (yellow star), and an atypical pigment network (black arrow).

**Table 1 cancers-17-03859-t001:** Clinical and histopathological characteristics of NAM and DNM.

Variable	NAM	DNM
Mean age (years)	52.0	54.0
Female (%)	42.0	48.0
Male (%)	58.0	52.0
Trunk (%)	65.6	51.7
Lower limbs (%)	14.4	22.6
Upper limbs (%)	13.3	16.7
Head and neck (%)	4.4	7.3
Acral (%)	2.2	1.7
Mean Breslow (mm)	0.55	0.84
Mean mitoses (/mm^2^)	0.17	1.21
Ulceration (%)	2.2	9.4
SLN positivity (%)	1.1	6.3
Distant metastases (%)	0.0	6.6
Melanoma-specific mortality (%)	0.0	0.69

**Table 2 cancers-17-03859-t002:** Univariate logistic regression analysis assessing the association between patient/tumor features and melanoma subtype. Breslow thickness and mitotic rate were the only variables significantly associated with NAM versus DNM.

Patient Features	*p*-Value	Odds Ratio (95% CI)
Gender	0.935	0.96 (0.36–2.54)
Age	0.542	1.10 (0.98–1.05)
Breslow thickness	0.037	1.008 (1.00–1.20)
Mitoses	0.009	0.38 (0.19–0.79)
Positive sentinel lymph node	0.671	0.60 (0.06–6.50)
Ulceration	0.728	1.42 (0.20–10.10)
Distant Metastasis	0.998	0.00 (0–0)
Mortality	1.0	0.20 (0–0)

## Data Availability

Data is contained within the article.

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
