# Peer review of "Nevus-Associated and De Novo Melanoma: A Cross-Sectional Study on Prognostic Differences"

_cancers, 2025, doi:10.3390/cancers17233859_

Round 1

Reviewer 1 Report

Comments and Suggestions for Authors

Comments to the Authors:

Nevus-associated melanoma represents a subtype of melanoma that garners significant attention from dermatologists in clinical practice. Therefore, we are particularly intrigued by research investigating the differences between nevus-associated melanomas and melanomas that arise de novo.

The following points, if addressed, would strengthen the manuscript and enhance its relevance for clinicians:

  1. The variable listed in the Materials and Methods is "metastasis," while the independent variable presented in Table 1 is "Distant metastases." It should be clarified whether "metastasis" in the Materials and Methods specifically refers to distant metastasis, excluding sentinel lymph node involvement. We recommend providing a precise definition of "metastasis" as used in this study and ensuring consistent terminology is applied throughout the manuscript.
  2. The analysis employs melanoma subtype as the dependent variable in a logistic regression with factors like Breslow thickness to explore their relationship. Employing "Metastasis" and "Mortality" as dependent variables is conceptually problematic, as it does not align with the presumed biological and temporal sequence. It is more logical to investigate whether the melanoma subtype (the exposure) influences the risk of metastasis or mortality (the outcomes), not the reverse. A more robust analytical approach would be to model "Metastasis" as the dependent variable, with melanoma subtype as an independent variable. This would allow for an investigation into the effect of melanoma subtype on metastasis risk. Furthermore, sequentially adjusting for other covariates could help determine whether melanoma subtype exerts an independent influence on metastasis.

3. Regarding "Mortality," it is recommended to perform survival analysis. This would involve investigating the impact of melanoma subtype on survival outcomes after adjusting for relevant covariates. Standard survival analysis methods, such as Kaplan-Meier curves and Cox proportional hazards regression, would be appropriate for this purpose.

Author Response

  • Comment 1: [The variable listed in the Materials and Methods is "metastasis," while the independent variable presented in Table 1 is "Distant metastases." It should be clarified whether "metastasis" in the Materials and Methods specifically refers to distant metastasis, excluding sentinel lymph node involvement. We recommend providing a precise definition of "metastasis" as used in this study and ensuring consistent terminology is applied throughout the manuscript.]

Response: Thank you for pointing this out. We agree that the terminology required clarification. In our study, the variable “metastasis” refers exclusively to distant metastasis and does not include sentinel lymph node involvement, which was recorded as a separate variable. We have now revised the Materials and Methods to specify this clearly and to ensure that all variables collected in the study are fully listed and consistently defined. Terminology has also been aligned throughout the manuscript, including in Table 2. The corresponding changes can be found in the revised version on page 3, lines 112, 121-122 and in Table 2 on page 4.

  • Comment 2: [The analysis employs melanoma subtype as the dependent variable in a logistic regression with factors like Breslow thickness to explore their relationship. Employing "Metastasis" and "Mortality" as dependent variables is conceptually problematic, as it does not align with the presumed biological and temporal sequence. It is more logical to investigate whether the melanoma subtype (the exposure) influences the risk of metastasis or mortality (the outcomes), not the reverse. A more robust analytical approach would be to model "Metastasis" as the dependent variable, with melanoma subtype as an independent variable. This would allow for an investigation into the effect of melanoma subtype on metastasis risk. Furthermore, sequentially adjusting for other covariates could help determine whether melanoma subtype exerts an independent influence on metastasis.]

Response: Thank you for this important methodological comment. We fully agree that metastasis and mortality represent clinical outcomes and should not be modeled as predictors of melanoma subtype. Conceptually, the appropriate analytical framework would be to examine metastasis or mortality as dependent variables, with melanoma subtype as the exposure, and to sequentially adjust for established prognostic factors.

However, in our cohort the number of metastatic (n = 19) and melanoma-specific fatal events (n = 2) was extremely low. This event rate is insufficient to support reliable outcome-based modeling, such as logistic regression or Cox proportional hazards analysis, which would produce unstable estimates and violate model assumptions. For this reason, we restricted multivariable analysis to predictors of subtype classification (NAM vs. DNM), where sample size allowed for valid model performance.

We have now clarified these considerations in the revised manuscript. The corresponding revisions can be found in the Limitations section on page 8 (lines 344–350).

  • Comment 3: [Regarding "Mortality," it is recommended to perform survival analysis. This would involve investigating the impact of melanoma subtype on survival outcomes after adjusting for relevant covariates. Standard survival analysis methods, such as Kaplan-Meier curves and Cox proportional hazards regression, would be appropriate for this purpose.]

Response: Thank you for this valuable suggestion. We fully agree that Kaplan–Meier survival curves and Cox proportional hazards regression would be the appropriate analytical approach to evaluate the impact of melanoma subtype on mortality after adjustment for relevant covariates. However, in our cohort melanoma-specific deaths were extremely rare (n = 2). This event rate is far below the minimum required to generate meaningful Kaplan–Meier estimates or to fit a Cox model without violating proportional hazards assumptions, producing unstable estimates, or inducing model overfitting.

For these reasons, a survival analysis could not be reliably performed. We have now clarified this explicitly in the revised manuscript, noting that the very low number of fatal events precludes survival modeling and limits the ability to draw conclusions regarding long-term prognosis. We also emphasize that larger multicenter datasets will be necessary to determine whether melanoma subtype exerts an independent influence on survival outcomes. The corresponding clarification has been added in the Limitations section (page 8, lines 344–346).

Reviewer 2 Report

Comments and Suggestions for Authors

Dear authors,

Congratulations on your well done paper and very interesting conclusions. I read your manuscript with interest.

-Please mark the dermoscopic features seen throughout the images with an arrows/stars/etc

-The trunk was more frequent location in both group, could you also please add the information whether there were any differences between genders?

-I enjoyed reading the discussion which is very comprehensive and matches my thoughts on the topic

Other than that, I do not have any other remarks.

Author Response

  • Comment 1: [Please mark the dermoscopic features seen throughout the images with an arrows/stars/etc.]

Response: Thank you for this suggestion. We have now annotated all dermoscopic images with arrows and asterisks to clearly indicate the corresponding features described (e.g., regression areas, atypical pigment network, blue-white veil). In addition, all figure captions have been updated accordingly to provide clear and consistent descriptions of the marked structures. These changes are included in the revised version of the manuscript.

  • Comment 2: [The trunk was more frequent location in both group, could you also please add the information whether there were any differences between genders?]

Response: Thank you for this insightful comment. We reviewed our dataset to evaluate whether anatomical site distribution differed between males and females in either the NAM or DNM cohorts. No statistically significant sex-related differences were observed. In both men and women, the trunk was the most common site of involvement, followed by the lower and upper extremities. Likewise, no differences by sex were found in age at diagnosis, histopathologic parameters (Breslow thickness, mitotic rate, ulceration), sentinel lymph node status, distant metastasis, or melanoma-specific mortality. We have now added a clarifying sentence in the Results section on page 3 (lines 133–136).

Reviewer 3 Report

Comments and Suggestions for Authors

This is a retrospective analysis of clinical features of NAM and DNM. The finding reveals that NAM has a lower thickness and lower mitotic activity compared to DNM. However, the addictive value of the study is limited as an observational study. It will be helpful to connect these findings with prognosis (although no difference in this study) and treatment response.

Author Response

  • Comment 1: [This is a retrospective analysis of clinical features of NAM and DNM. The finding reveals that NAM has a lower thickness and lower mitotic activity compared to DNM. However, the addictive value of the study is limited as an observational study. It will be helpful to connect these findings with prognosis (although no difference in this study) and treatment response.]

Response: Thank you for this constructive comment. Although our cohort did not show differences in melanoma-specific mortality or sentinel node involvement between NAM and DNM, we agree that connecting our findings to the broader prognostic and therapeutic implications enhances the impact of the study. We have now expanded the Discussion to contextualize the observed differences in Breslow thickness, mitotic activity, and ulceration within current prognostic models. Furthermore, we incorporated a paragraph discussing how NAM and DNM may differ biologically in ways that could influence treatment response, particularly in the context of BRAF-mutated disease and patterns of cumulative sun damage. These additions reinforce the clinical relevance of our findings and align the observational data with meaningful prognostic and therapeutic considerations. This change can be found on page 8 lines 321-341.

Reviewer 4 Report

Comments and Suggestions for Authors

The manuscript is concise and well written , but authors should remember that melanoma is not a cancer.Melanoma is a tumor. Such term should be changed in the text.

Author Response

  • Comment 1: [The manuscript is concise and well written , but authors should remember that melanoma is not a cancer.Melanoma is a tumor. Such term should be changed in the text.]

Response: Thank you for this remark. We agree with the reviewer’s observation that melanoma should be referred to as a tumor rather than a cancer. We have carefully revised the manuscript and replaced all occurrences of “melanoma cancer” or “cancer” (when referring specifically to melanoma) with the correct terminology (“melanoma” or “tumor,” as appropriate). The corresponding changes can be found in the revised manuscript on page 1, lines 13–15.